# Continuous glucose monitoring in obese pregnant women with no hyperglycemia on glucose tolerance test

**Rosa Maria Rahmi**[1☉]*, **Priscila de Oliveira**[2☉], **Luciano Selistre**[3‡], **Paulo Cury Rezende**[4‡], **Gabriela Neuvald Pezzella**[5‡], **Pâmela Antoniazzi dos Santos**[6‡], **Daiane de Oliveira Pereira Vergani**[7‡], **Sônia Regina Cabral Madi**[8‡], **José Mauro Madi**[8☉]

1 Department of Endocrinology, General Hospital of Caxias do Sul, Caxias do Sul University, Caxias do Sul, RS, Brazil, 2 Master Degree Student in Health Sciences, Caxias do Sul University, Caxias do Sul, RS, Brazil, 3 Department of Nephrology, General Hospital of Caxias do Sul, Caxias do Sul, RS, Brazil, 4 Department of Atherosclerosis, Heart Institute of the University of São Paulo Medical School, São Paulo, SP, Brazil, 5 Institutional Scientific Initiation Scholarship Program of Caxias do Sul University, Caxias do Sul, RS, Brazil, 6 Department of Nutrition, Caxias do Sul University, Caxias do Sul, RS, Brazil, 7 Department of Nursing, Caxias do Sul University, Caxias do Sul, RS, Brazil, 8 Department of Obstetrics and Gynecology, General Hospital of Caxias do Sul, Caxias do Sul University, Caxias do Sul, RS, Brazil

☉ These authors contributed equally to this work.
‡ These authors also contributed equally to this work.
* rmrgarcia@ucs.br

**Data Availability Statement:** The authors confirm that the data available at the provided DOI contains

## Abstract

### Objective

The objective of the present study was to compare 24-hour glycemic levels between obese pregnant women with normal glucose tolerance and non-obese pregnant women.

### Methods

In the present observational, longitudinal study, continuous glucose monitoring was performed in obese pregnant women with normal oral glucose tolerance test with 75 g of glucose between the 24th and the 28th gestational weeks. The control group (CG) consisted of pregnant women with normal weight who were selected by matching the maternal age and parity with the same characteristics of the obese group (OG). Glucose measurements were obtained during 72 hours.

### Results

Both the groups were balanced in terms of baseline characteristics (age: 33.5 [28.7–36.0] vs. 32.0 [26.0–34.5] years, p = 0.5 and length of pregnancy: 25.0 [24.0–25.0] vs. 25.5 [24.0–28.0] weeks, p = 0.6 in the CG and in the OG, respectively). Pre-breakfast glycemic levels were 77.77 ± 10.55 mg/dL in the CG and 82.02 ± 11.06 mg/dL in the OG (p<0.01). Glycemic levels at 2 hours after breakfast were 87.31 ± 13.10 mg/dL in the CG and 93.48 ± 18.74 mg/dL in the OG (p<0.001). Daytime blood glucose levels were 87.6 ± 15.4 vs. 93.1 ± 18.3 mg/dL (p<0.001) and nighttime blood glucose levels were 79.3 ± 15.8 vs. 84.7 ± 16.3 mg/dL (p<0.001) in the CG and in the OG, respectively. The 24-hour, daytime, and nighttime values of the area under the curve were higher in the OG when compared

minimal data set which is consisted by the data set used to reach our conclusions drawn in the manuscript with related metadata and methods, and to replicate the reported study findings in their entirety. Data can be accessed through in Harvard Dataverse, https://doi.org/10.7910/DVN/2KZXMJ.

**Funding:** The authors received no specific funding for this work.

**Competing interests:** The authors have declared that no competing interests exist.

with the CG (85.1 ± 0.16 vs. 87.9 ± 0.12, 65.6 ± 0.14 vs. 67.5 ± 0.10, 19.5 ± 0.07 vs. 20.4 ± 0.05, respectively; p<0.001).

## Conclusion

The results of the present study showed that obesity in pregnancy was associated with higher glycemic levels even in the presence of normal findings on glucose tolerance test.

## Introduction

During the last four decades, prevalence of obesity has increased dramatically around the world. In 2016, the World Health Organization (WHO) estimated that approximately 650 million adults were obese, representing approximately 13% of the world's adult population. Obesity affects all age groups and both sexes irrespective of the income levels [1]. Concomitant with the global increase in obesity, the number of obese pregnant women has also increased [2].

The association of obesity with pregnancy has been an important public health problem and a major challenge for the professional team responsible for assisting this population. Maternal obesity is associated with adverse pregnancy and perinatal outcomes and long-term complications related to maternal and fetal health [3]. Current evidences support the strong association between obesity and gestational diabetes mellitus (GDM) [4, 5]. Excess fat tissue releases increased amounts of unesterified fatty acids, glycerol, hormones, pro-inflammatory cytokines, and other factors that participate in the development of insulin resistance (IR). IR and dysfunctional beta-pancreatic cells are the main factors causing hyperglycemia [6, 7]. In this context, maternal obesity causes imbalance in glycemic homeostasis during pregnancy, resulting in an increased risk of GDM [8].

Screening and diagnosis of GDM has improved in recent decades. However, there is still a lack of universally accepted consensus [9–11]. In 2010, the International Association of Diabetes in Pregnancy Study Group (IADPSG) [12] updated the diagnostic criteria based on the results of an important study, namely the Hyperglycemia and Adverse Pregnancy Outcomes (HAPO) study (13). These criteria were widely accepted by national and international organizations. The HAPO study suggested a strong and continuous relationship between maternal blood glucose and adverse outcomes [13]. The study proposed a lower glycemic threshold to detect GDM compared to other international guidelines [9, 14–16].

GDM is mainly diagnosed using the oral glucose tolerance test (OGTT), which is based on a limited number of plasma glucose level readings after glucose overload [16].

After diagnosis, GDM needs to be treated by a multidisciplinary team. Glycemic control supervised by glycemic self-monitoring at specific time points (especially preprandial and postprandial readings) is crucial to reduce the risk of adverse maternal and fetal outcomes [17]. During pregnancy, the proposed range of glycemic levels to manage hyperglycemia is more limited. This rigor is believed to positively influence the adverse perinatal outcomes. However, such monitoring is based on a limited number of analyses within 24 hours and long periods between meals are not monitored.

Maternal blood glucose has a dynamic variation within 24 hours and is influenced by numerous factors such as insulin sensitivity, diet, lifestyle, stress, sleep, and others [18, 19].

Currently, with technological developments in continuous glucose monitoring (CGM), it is possible to assess daily glycemic fluctuations with greater accuracy. Several studies have been designed to allow better understanding of the effect of hyperglycemia on the temporal behavior

of glycemic levels in pregnancy [20–23]. However, very few studies have analyzed the continuous evolution of glycemic levels during the period in pregnancy without glucose intolerance [24–26].

Obese women with presumably normal glucose tolerance may experience adverse perinatal complications similar to those observed in women with GDM [4, 27]. Although the HAPO trial have demonstrated that lower glucose values than those currently adopted to diagnose gestational diabetes resulted in improvement of adverse outcomes [13], the debate continues over what constitutes normoglycemia in pregnancy. Furthermore, there are few data on the glycemic patterns during 24 hours in obese pregnant women without GDM.

Thus, the present study was designed to assess the 24-hour glycemic profile using continuous glucose monitoring.in obese and not obese pregnant women, without glucose intolerance according to the criteria proposed by the IADPSG [12]. We postulated that 24-h glucose measures (24-h area under the curve [AUC]) would be higher among obese women than among non-obese women.

## Materials and methods

The present prospective, observational, longitudinal study involving pregnant women was followed up by the Obstetrics and Gynecology Service of the General Hospital of the University of Caxias do Sul, RS, Brazil. The study was approved by the Ethics and Human Resources Committee of the University of Caxias do Sul. It was conducted according to the ethical principles of the Declaration of Helsinki. All participants signed an Informed Consent Form.

The study was conducted from June 2018 to July 2019. We recruited outpatient pregnant woman underwent OGTT with 75 g of glucose between the 24[th] and the 28[th] gestational weeks. We included women with fasting glycemic levels below 92 mg/dL (5.1 mmol/L), 1-hour glycemic levels below 180 mg/dL (10.0 mmol/L), and 2-hour glycemic levels below 153 mg/dL (8.5 mmol/L). Only pregnant women with gestational age between 24 to 32 weeks and aged 18 to 35 years were included. The defined age range was proposed to reduce the impact of advancing age on GDM risk." The exclusion criteria were twin pregnancy; fetal malformation; pregnant women with uncontrolled chronic diseases; smoking; alcoholism; and use of corticosteroids, beta-blockers, or hyperglycemic drugs. Ten pregnant women with pre-gestational obesity (body mass index [BMI] range: 30–40 kg/m$^2$) were consecutively included to compose the obese group (OG). Another 10 women with normal pre-pregnancy weight (BMI range: 18.5–24.9 kg/m$^2$) matched (1:1) by maternal age, parity, and length of pregnancy were selected to control group (CG).

In order to obtaining the data in a real-life context, all pregnant women were continuously monitored by the prenatal care team without any interference or request from the researchers. The following data were collected from the medical records immediately after OGTT: age, pregestational BMI, parity, weight gain during pregnancy, gestational age at the time of OGTT, OGTT results (fasting, at 1 hour after overload, and at 2 hours after overload), family history of cardiovascular disease, and family history of diabetes. Pregestational BMI was calculated according to the WHO standards and expressed as weight (kg)/height (m)$^2$. Maternal weight gain during pregnancy was calculated by subtracting the body weight at the time of OGTT from the pre-pregnancy weight. Interstitial glucose profiles were measured immediately after inclusion.

### Continuous glucose monitoring

A CGM system iPro™2 Professional CGM, by Medtronic Principal Executive Office 20 Lower Hatch Street Dublin 2, Ireland), was used to measure interstitial glucose concentrations over a

period of 24 hours for 3 consecutive days. The sensors were inserted in the subcutaneous tissue in the lower abdomen on the right or the left side. The sensors were connected to the transmitters attached to the skin. The sensor recorded approximately 288 blood glucose level readings in each pregnant woman over 24 hours. After 72 hours, the data were stored in a database. The monitors were calibrated by inserting capillary blood glucose level measured three times a day (preprandial measurements) using the Accu-Chek Active® device (Roche, Basel, Switzerland). Concomitantly, the women were requested to record the time at the start of the main meals and the time at the start of physical exercise.

The sample size was determined such that the width of the two-sided 95% confidence interval for the between-group difference, under an assumption of normally distributed data for the change in glucose level from baseline to 24 hours, was 0.5 percentage points. We estimated that 10 participants per group with 30 measurements would provide the study with 95% power to detect a prespecified effect (standard deviation) of 5 mg/dL on the primary outcome, assuming a two-sided type I error of 1%. To compute the Cohen's effect size for the Pillai statistic from mean and variance-covariance matrix and, as input method, standard deviation and correlation matrix. We get a value 0.801 for Pillai's V and the Cohen's effect size = 0.341. The parameters were lambda = 40.50, values critical F = 11.26. A total of 16 patients, with 8 in each treatment group, would meet this requirement. The data were expressed as mean ± standard deviation, median [interquartile range], and percentage. Exploratory analysis of the descriptive data was performed using Student's t-test, Wilcoxon-Mann-Whitney test, and Pearson's chi-squared test. Since blood glucose concentrations of nestlings from the same brood are not independent, the glucose concentrations were analyzed using mixed linear models with brood identity included as a random controlling factor. In the first step, the glucose levels were modeled according to a linear mixed model with random intercept to quantify the effect of the group (obese or non-obese). The mean values of the two groups were compared using t-test in the linear mixed model. In the second step, two models were built: a first model that included variables "group" and "time" and a second model that included an interaction between the variables "group" and "time." The second model allowed quantification of the change in the effect of the group type according to time. Analysis of variance was used to compare the two nested models and to determine the statistical significance of the interaction. The models were adjusted by the restricted maximum likelihood method using the LME function of the NLME package. Tukey's post hoc test was used for multiple comparisons. The analyses were performed using R for Windows, version 3.1.1 (R-Cran project, http://cran.r-project.org/, The R foundation, Vienna, Austria). Nominal p-values <0.05 were considered statistically significant.

## Results

Altogether, 20 pregnant women were included and evaluated in this study. The baseline characteristics of the population in the OG (n = 10) and in the CG (n = 10) are described in Table 1. The median maternal age was 33.5 [28.7–36.0] years in the CG and 32.0 [26.0–34.5] years in the OG (p = 0.5). The pregestational BMI (kg/m$^2$) was 22.1 [21.7–23.8] in the CG and 39.9 [35.8–41.9] in the OG (p<0.001). Maternal weight gain until the day of OGTT tended to be greater in the OG (8.0 [5.5–10.7] kg) than in the CG (2.6 [0.00–8.6] kg) (p = 0.09). The analysis of OGTT results revealed that the fasting glycemic levels tended to be higher in the OG (75.5 [72.0–79.7] mg/dL) than in the CG (81.5 [74.2–87.0] mg/dL) (p = 0.08). Blood glucose levels at 1 and 2 hours after glucose overload showed no significant differences between the groups. Moreover, no statistically significant difference was observed in parity and in family history of cardiovascular disease and diabetes between the groups (Table 1).

**Table 1. Characteristics of pregnant women in the obese and control groups.**

|  | CG (n = 10) | OG (n = 10) | p-value |
|---|---|---|---|
| Age (years). | 33.50 [28.75–36.00] | 32.0 [26.0–34.5] | 0.5 |
| Parity $\geq$ 1(n) | 9 | 10 | 1.0 |
| Pregestational BMI (kg/m$^2$) | 22.15 [21.70–23.82] | 39.95 [35.85–41.88] | <0.001 |
| Weight gain (kg) | 2.65 [0.00–8.57] | 8.00 [5.50–10.75] | 0.09 |
| Family history of CVD (%) | 30 | 20 | 1.00 |
| Family history of diabetes (%) | 40 | 50 | 1.00 |
| Length of pregnancy (weeks)[a] | 25.0 [24.0–25.0] | 25.5 [24.0–28.0] | 0.6 |
| OGTT (mg/dL) |  |  |  |
| Fasting | 75.50 [72.00–79.75] | 81.50 [74.25–87.00] | 0.08 |
| 1 hour | 129.0 [117.0–141.0] | 134.0 [120.0–161.0] | 0.4 |
| 2 hours | 110.00 [95.25–116.00] | 109.00 [93.75–124.50] | 0.9 |

[a] Length of pregnancy at the time of oral glucose tolerance test, OG: obese group, CG: control group, BMI: body mass index; OGTT: oral glucose tolerance test; wk: week; CVD: cardiovascular disease. Data are medians, Interquartile range (IQR), and percentage. P-values were calculated using by Wilcoxon-Mann-Whitney test and chi-squared test.

The CGM data of pregnant women from both the groups are presented in Table 2. A significant difference was observed in blood glucose levels before (77.77 mg/dl ± 10.55 vs. 82.02 ± 11.06, p<0.01) and 2 hours after breakfast (87.31 mg/dl ± 13.10 vs. 93.48 ± 18.74, p<0.001) between the CG and the OG. No significant difference was observed in the values within 1 hour after breakfast. No significant differences were observed in glucose levels before and after lunch and dinner between the groups. Additionally, blood glucose levels during the day (between 6:00 am and 12:00 pm) were significantly higher in the OG compared to those in the CG (93.08 mg/dl ± 18.30 vs. 87.58 ± 15.40, p<0.001). Similarly, blood glucose levels at night (between 12:00 pm and 6:00 am) were significantly higher in the OG compared to those in the CG (84.73 mg/dl ± 16.31 vs. 79.35 ± 15.76, p<0.001).

The areas under the curve (AUCs) for blood glucose levels during the day and at night were 67.47 mg/dl ± 0.105 and 20.42 ± 0.05, respectively in the OG and 65.56 mg/dl ± 0.144 and 19.53 ± 0.072, respectively in the CG (p<0.001) (Table 2). The 24-hour AUC for blood glucose levels was 85.08 mg/dl ± 0.161 in the OG and 87.89 ± 0.116 in the CG (p<0.001) (Table 2 and Fig 1).

Table 3 shows the isolated effect of obesity on longitudinal blood glucose variation. This effect was significant at night (78.10 mg/dl [95% confidence interval: 72.61–83.60] in the CG vs. 82.78 mg/dl [95% confidence interval: 78.60–86.96] in the OG, p<0.001).

## Discussion

The present study clearly showed a difference in temporal evolution of glycemic levels between obese and non-obese pregnant women without hyperglycemia according to the IADPSG criteria [12]. The national protocol in Brazil suggests that GDM screening should be performed using OGTT with 75 g of glucose between the 24[th] and the 28[th] gestational weeks in pregnant women with no previous glycemic changes. GDM is diagnosed when the following levels were reached or exceeded: fasting glucose level of 92 mg/dl, 1-hour level of 180 mg/dL, and 2-hour level of 153 mg/dL [16]. In the studied population, the analysis of blood glucose levels at fasting, at 1 hour, and at 2 hours after 75 g glucose overload confirmed that none of the pregnant women met or exceeded these criteria. However, fasting glycemic levels in the OG tended to be higher than those in the CG (p = 0.08) at the time of screening. None of the pregnant

**Table 2. Continuous glucose monitoring data in control and obese groups.**

|  | CG | OG | p-value* |
|---|---|---|---|
| Glucose (mg/dL) |  |  |  |
| Before breakfast | 77.77 ± 10.55 | 82.02 ± 11.06 | <0.01 |
| 1 hour after breakfast | 94.25 ± 15.70 | 97.26 ± 11.06 | 0.8 |
| 2 hours after breakfast | 87.31 ± 13.10 | 93.48 ± 18.74 | <0.001 |
| Before lunch | 82.77 ± 15.15 | 85.26 ± 15.65 | 0.2 |
| 1 hour after lunch | 97.74 ± 13.60 | 97.71 ± 14.96 | 0.6 |
| 2 hours after lunch | 93.78 ± 12.30 | 91.13 ± 13.65 | 0.15 |
| Before dinner | 82.80 ± 2.75 | 86.68 ± 2.04 | 0.08 |
| 1 hour after dinner | 94.42 ± 19.05 | 94.02 ± 17.35 | 0.8 |
| 2 hours after dinner | 90.65 ± 23.37 | 92.78 ± 20.27 | 0.2 |
| Daytime | 87.58 ± 15.40 | 93.08 ± 18.30 | <0.001 |
| Nighttime | 79.35 ± 15.76 | 84.73 ± 16.31 | <0.001 |
| AUC (mg/min/dL) |  |  |  |
| Day | 65.56 ± 0.144 | 67.47 ± 0.105 | <0.001 |
| Night | 19.53 ± 0.072 | 20.42 ± 0.05 | <0.001 |
| 24 hours | 85.08 ± 0.161 | 87.89 ± 0.116 | <0.001 |

CG: control group, OG: obese group, AUC: area under the curve. Preprandial and postprandial glucose level is the mean of three consecutive values before or after the respective meal. Daytime glucose is the mean glucose level between 6:00 am and 12:00 pm. Nighttime glucose is the mean glucose level between 12:00 pm and 6:00 am. Daytime AUC is between 6:00 am and 12:00 pm and nighttime AUC is between 12:00 pm and 6:00 am.

*The p-values (obese vs. control) are based on F statistics for comparisons test.

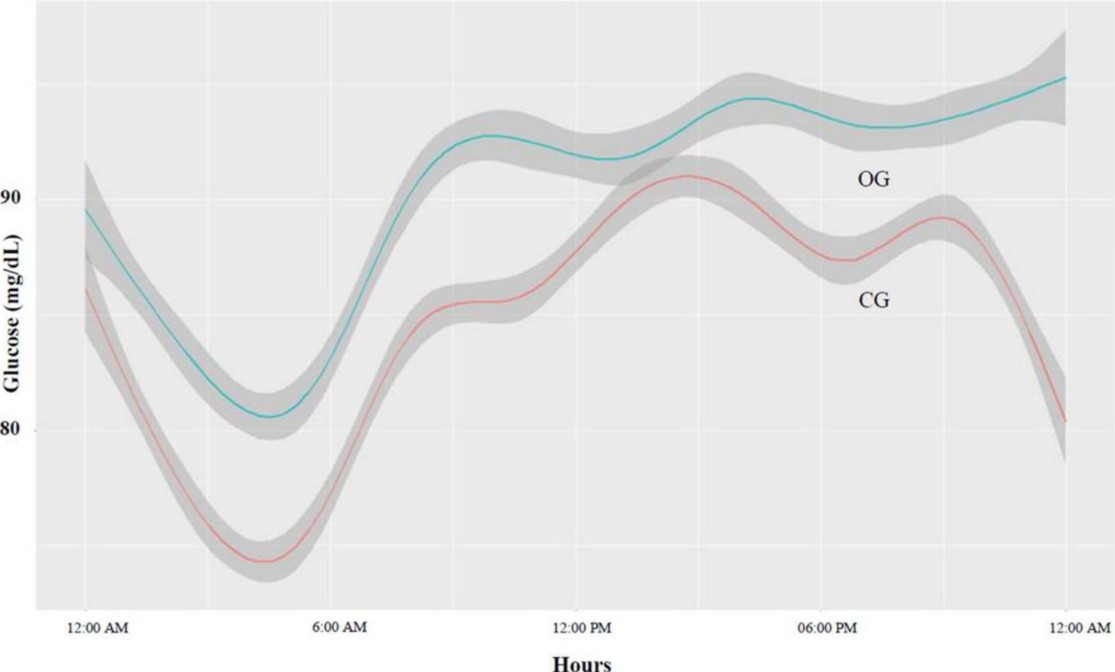

**Fig 1. Glucose profile during 24 hours in the obese and control group.** Obese group (OG) is represented by the green smooth curve (lambda = 1,000,000) and control group (CG) by the red smooth curve.

**Table 3. The mixed linear model to analyze the effect of obesity on the glucose levels.**

|  | CG (95.0% CI) | OG (95.0% CI) | p-value |
|---|---|---|---|
| Whole sample | 84.94 (81.55; 88.33) | 88.58 (85.43; 91.63) | 0.17 |
| Daytime | 86.87 (82.90; 90.84) | 90.21 (87.20; 93.24) | 0.25 |
| Nighttime | 78.10 (72.61; 83.60) | 82.78 (78.60; 86.96) | <0.001 |

CG: control group, OG: obese group, CI: confidence interval.

women in the study exhibited evidence of hyperglycemia. Therefore, they were routinely monitored without strict blood glucose level control until the end of pregnancy. No intervention was performed by the researchers. The objective of this study was to assess blood glucose levels without changing the routine in a population at high risk for metabolic diseases.

The obese pregnant women in the present study were referred to a reference center for high-risk pregnancies at the General Hospital of Caxias do Sul. There has been a significant increase in the number of women with severe obesity in recent years due to the global obesity epidemic that also affects women of reproductive age [2]. According to the study by Kim et al., the rate of GDM in a population with severe obesity (35–64.9 kg/m$^2$) was 11.5% and the relative risk of GDM was 5.0 (95% confidence interval: 3.6–6.9) even after adjustment for maternal age, race/ethnicity, parity, and marital status [8]. In addition to pregestational BMI, weight gain during pregnancy may also be associated with an increased risk for GDM [28, 29]. In the present study population, weight gain during the study period was higher in the OG (median: 8.00 kg) when compared with that in the CG (median: 2.65 kg), which is an additional factor for increased risk of hyperglycemia. Despite the high pregestational BMI and the greater weight gain in obese pregnant women, GDM was not detected at the time of screening. Thus, there is a possibility of dysglycemia in later stages of pregnancy in risk groups with a negative GDM test. Gomes et al. showed that among 448 obese pregnant women with a negative GDM test, 30.1% (n = 135) exhibited dysglycemia at the end of the third trimester, as assessed by increased hemoglobin A1c levels [30]. A secondary analysis of the HAPO study in a population of 23,316 pregnant women showed that 2,247 (9.6%) women were obese without a diagnosis of hyperglycemia and this condition showed an independent association with fetal hyperinsulinemia, growth, and adiposity, similar to the outcomes observed in GDM [4]. This subject continues being discussed due to the scarce literature on the effects of late glycemic changes and maternal lipid profile [31, 32] on perinatal outcomes.

Blood glucose levels at specific time points (2 hours before and 2 hours after breakfast) were significantly higher in the OG. However, the levels did not exceed the recommended limits for these time points (<95 and <120 mg/dl, respectively) [17]. These are the recommended time points to monitor pregnant women with hyperglycemia. At these time points, blood glucose levels remained within the presumably normal range in both the groups. Harmon reported significant differences in glycemic levels at 1 and 2 hours after meals [26]. Stratified analysis by pregestational maternal weight conducted by Yogev et al. showed that preprandial, 1-hour postprandial, and 2-hour postprandial glycemic levels were significantly higher in obese pregnant women [25].

A detailed analysis of blood glucose samples repeated for 72 hours showed higher fluctuation in obese pregnant women than in non-obese pregnant women (assessed by the AUC). Similar behavior was observed when the analysis was divided into two periods (day and night). In addition, obesity was associated with a higher mean blood glucose at night. These data suggest that fetuses of the women from the OG could potentially be exposed to a blood glucose pattern that is higher than normal. These findings are consistent with the findings of Harmon

et al. [26] who evaluated groups of pregnant women without hyperglycemia with and without dietary interference and reported that the AUC was always higher in obese pregnant women regardless of dietary control. In the present study, the OG included pregnant women with more severe obesity (median BMI: 39.95) and the criteria for excluding glucose intolerance in the population were different. However, Yogev et al. [25] showed that obese women exhibited significantly lower mean glucose levels at night compared to non-obese women.

Differences in glycemic homeostasis between obese and non-obese pregnant women were didactically presented by analyzing temporal blood glucose variations over long periods, which is possible only with the CGM systems.

Despite the few studies available in the literature, the following questions should be discussed. 1) Should the glycemic targets for obese pregnant women be individualized? 2) Could the nocturnal glycemic changes be related to increased fetal fat and/or macrosomia in obese women without GDM?

Increasing maternal obesity rates have challenged researchers to characterize the metabolic profile of this population in a better way. Glycemic control is not adequately addressed during the follow-up in most of the obese pregnant women without GDM. On the other hand, glucose self-monitoring has limitations, as it does not include the night period. The present study suggests the need for more evidence on glycemic targets in obese women during pregnancy. The sample size in the present study did not allow correlations with perinatal outcomes. However, the use of statistical modeling and the strict composition of the two groups clearly showed distinct behaviors in dynamic changes in blood glucose levels over long periods.

## Conclusion

"In conclusion, the present study demonstrated that continuously assessed blood glucose levels were higher in obese pregnant women without GDM than in non-obese pregnant women and this effect was more evident at night. Additional studies may lead to the better understanding of this metabolic alteration and its possible correlation with adverse neonatal outcomes."

## Acknowledgments

The authors are grateful to Hospital Geral de Caxias do Sul for receive this search.

## Author Contributions

**Conceptualization:** Rosa Maria Rahmi, José Mauro Madi.

**Data curation:** Rosa Maria Rahmi.

**Formal analysis:** Luciano Selistre.

**Investigation:** Priscila de Oliveira.

**Methodology:** Rosa Maria Rahmi, Priscila de Oliveira, José Mauro Madi.

**Resources:** Rosa Maria Rahmi, José Mauro Madi.

**Supervision:** Rosa Maria Rahmi, José Mauro Madi.

**Writing – original draft:** Rosa Maria Rahmi, Priscila de Oliveira.

**Writing – review & editing:** Rosa Maria Rahmi, Paulo Cury Rezende, Gabriela Neuvald Pezzella, Pâmela Antoniazzi dos Santos, Daiane de Oliveira Pereira Vergani, Sônia Regina Cabral Madi, José Mauro Madi.

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
