## [Decision Letter · Decision Letter 0]

30 Dec 2020

PONE-D-20-32574

Continuous glucose monitoring in obese pregnant women with no hyperglycemia on glucose tolerance test

PLOS ONE

Dear Dr. Vergani,

Thank you for submitting your manuscript to PLOS ONE. After careful consideration, we feel that it has merit but does not fully meet PLOS ONE’s publication criteria as it currently stands. Therefore, we invite you to submit a revised version of the manuscript that addresses the points raised during the review process.

We look forward to receiving your revised manuscript.

Kind regards,

Muhammad Sajid Hamid Akash

Academic Editor

PLOS ONE

Journal Requirements:

Reviewers' comments:

Reviewer's Responses to Questions

**Comments to the Author**

1. Is the manuscript technically sound, and do the data support the conclusions?

Reviewer #1: Yes

Reviewer #2: No

Reviewer #3: Partly

2. Has the statistical analysis been performed appropriately and rigorously? 

Reviewer #1: Yes

Reviewer #2: No

Reviewer #3: Yes

3. Have the authors made all data underlying the findings in their manuscript fully available?

Reviewer #1: Yes

Reviewer #2: Yes

Reviewer #3: No

4. Is the manuscript presented in an intelligible fashion and written in standard English?

Reviewer #1: Yes

Reviewer #2: No

Reviewer #3: Yes

5. Review Comments to the Author

Reviewer #1: Comments to the Author,

This manuscript nice presented the interesting finding that study the continuous glucose monitoring performed in obese pregnant women with normal oral glucose tolerance test with 75 g of glucose between the 24th and the 28th gestational weeks. Study was nicely designed to compare 24-hour glycemic levels between obese pregnant women with normal glucose tolerance and non-obese pregnant women. Overall, these findings are important and interesting. However, further improvement is necessary to solidify the Manuscript.

Here are a few comments and questions:

1. In current study author compare the obese (OG) And the (CG) between the 24th and the 28th gestational weeks. But every woman gain weight during pregnancy period. How Author compare between obese and control?

2. Pregnancy of each women i.e. 1st pregnancy, 2nd pregnancy or 3rd pregnancy data?

3. Diet plan absent in current study. Diet plan morning, lunch, dinner. Some food readily absorbed; Spicy food take more time to absorbed?

4. Time duration between breakfast, lunch and Dinner not mentioned?

5. There are some grammar errors. revise the manuscript.

Authors address these deficiencies, then the manuscript should be considered for publication.

Reviewer #2: The manuscript does not describe a technically sound piece of scientific research with data that supports the conclusions. Experiments have not been conducted rigorously, with appropriate controls, replication, and sample sizes

Reviewer #3: The authors consider age ranges between 18-35 for OG and CG but what is the influence of age, its not mention. Groups should be uniform w. r. t. age. The hypotheses at the end of the introduction are lacking in specificity and insufficiently motivated. Please ensure that your manuscript meets PLOS ONE's style requirements, including

those for file naming.

6. PLOS authors have the option to publish the peer review history of their article (what does this mean?). If published, this will include your full peer review and any attached files.

Reviewer #1: **Yes: **DR.MUHAMMAD TARIQ

Reviewer #2: **Yes: **Dr.Zunera Chauhdary

Reviewer #3: **Yes: **DR Shagufta Kamal

---

## [Author Response · Author response to Decision Letter 0]

15 Apr 2021

Reviewer #1

This manuscript nice presented the interesting finding that study the continuous glucose monitoring performed in obese pregnant women with normal oral glucose tolerance test with 75 g of glucose between the 24th and the 28th gestational weeks. Study was nicely designed to compare 24-hour glycemic levels between obese pregnant women with normal glucose tolerance and non-obese pregnant women. Overall, these findings are important and interesting. However, further improvement is necessary to solidify the Manuscript.

 Here are a few comments and questions:

1. In current study author compare the obese (OG) And the (CG) between the 24th and the 28th gestational weeks. But every woman gain weight during pregnancy period. How Author compare between obese and control?

Answer: The purpose of the study was to evaluate pregnant women with and without obesity at the beginning of pregnancy (pre-pregnancy). Although the obese group started their pregnancy with obesity and gained weight during the study period, they did not show glucose intolerance between the 24th and the 28th gestational weeks. The pregestational BMI (kg/m²) in control group was 22.1 [21.7–23.8] and weight gain until the day of Oral Glucose Tolerance Test was [2.6 (0.00–8.6) kg], therefore they were not obese at the time of inclusion. Moreover, maternal weight gain until the day of OGTT tended to be greater in the Obese Group [8.0 (5.5–10.7) kg] than in the Control Group [2.6 (0.00–8.6) kg], but with no statistical difference (p=0.09).

In the “Discussion” section, line 230 this was previously addressed: “In the present study population, weight gain during the study period was higher in the OG (median: 8.00 kg) when compared with that in the CG (median: 2.65 kg), which is an additional factor for increased risk of hyperglycemia. Despite the high pregestational BMI and the greater weight gain in obese pregnant women, GDM was not detected at the time of screening.” 

2. Pregnancy of each women i.e. 1st pregnancy, 2nd pregnancy or 3rd pregnancy data?

Answer: Most of the pregnant women in the study had their previous prenatal care followed up at another service, so I can not provide the 1st pregnancy, 2nd pregnancy or 3rd pregnancy data. However, the groups were matched by parity (number of children born). Although controversial, it has been hypothesized that the effect exerted by repeated pregnancies may induce a progressive increase in insulin resistance which, step by step, can subsequently facilitate the appearance of impaired glucose tolerance or diabetes or gestational diabetes (1). Our results showed that both groups were homogeneous according to parity (Table 1. Maternal characteristics of the studied patients)

3. Diet plan absent in current study. Diet plan morning, lunch, dinner. Some food readily absorbed; Spicy food take more time to absorbed?

Answer: Excellent question. In fact, the glycemic index of foods influences the daily glycemic excursion. Although all pregnant women received a food plan from the prenatal care team, our proposal was to evaluate the glycemic profile without any interference from the researchers, in order to obtaining the data in a real-life context; therefore this information was inserted in “Materials and methods” section, line 100: “In order to obtain the data in a real-life context, all pregnant women were continuously monitored by the prenatal care team without any interference or request from the researchers.”

4. Time duration between breakfast, lunch and Dinner not mentioned?

Answer: Good question. All pregnant women identified the time of the beginning of each meal (breakfast, lunch, and dinner), so we could check the values of pre- and post-prandial blood glucose. However, we did not analyze the time between the two, since this data does not contribute to the main objective, that was to show the glycemic fluctuation in the 24 hours of both groups and to analyze the difference of their respective AUC.

5. There are some grammar errors. revise the manuscript.

Answer: Thank you for this comment. The manuscript was previously reviewed by scientific editing service (Taylor & Francis Editing Services). The manuscript was revised again in order to correct any grammar errors.

Reviewer #2: 

1. Is the manuscript technically sound, and do the data support the conclusions?

The manuscript must describe a technically sound piece of scientific research with data that supports the conclusions. Experiments must have been conducted rigorously, with appropriate controls, replication, and sample sizes. The conclusions must be drawn appropriately based on the data presented. Reviewer #2: No

Answer: Excellent comments. We apologize for not describing the sample size calculation, which hampered your thorough review. This data was included in the “Material and Methods” section, line 121 (see below in red highlight), and certainly improved our manuscript 

“The sample size was determined such that the width of the two-sided 95% confidence interval for the between-group difference, under an assumption of normally distributed data for the change in glucose level from baseline to 24 hours, was 0.5 percentage points. We estimated that 10 participants per group with 30 measurements would provide the study with 95% power to detect a prespecified effect (standard deviation) of 5 mg/dL on the primary outcome, assuming a two-sided type I error of 1%. To compute the Cohen's effect size for the Pillai statistic from mean and variance-covariance matrix and, as input method, standard deviation and correlation matrix. We get a value 0.801 for Pillai’s V and the Cohen's effect size = 0.341. The parameters were lambda = 40.50, values critical F= 11.26. A total of 16 patients, with 8 in each treatment group, would meet this requirement.”

Despite a small sample size, it was sufficient to meet the main objective and to draw the conclusions. Moreover, in the “Material and Methods” section, line 87 (see below in red highlight), the paragraph on the study population was rewritten to highlight that the study was rigorously conducted, with appropriate controls, replication, and sufficient sample size. 

“We recruited outpatient pregnant woman underwent OGTT with 75 g of glucose between the 24th and the 28th gestational weeks. We included women with fasting glycemic levels below 92 mg/dL (5.1 mmol/L), 1-hour glycemic levels below 180 mg/dL (10.0 mmol/L), and 2-hour glycemic levels below 153 mg/dL (8.5 mmol/L). Only pregnant women with gestational age between 24 to 32 weeks and aged 18 to 35 years were included. The defined age range was proposed to reduce the impact of advancing age on GDM risk. The exclusion criteria were twin pregnancy; fetal malformation; pregnant women with uncontrolled chronic diseases; smoking; alcoholism; and use of corticosteroids, beta-blockers, or hyperglycemic drugs. Ten pregnant women with pre-gestational obesity (body mass index [BMI] range: 30–40 kg/m2) were consecutively included to compose the obese group (OG). Another 10 women with normal pre-pregnancy weight (BMI range: 18.5–24.9 kg/m2) matched (1:1) by maternal age, parity, and length of pregnancy were selected to control group (CG).

In order to obtaining the data in a real-life context, all pregnant women were continuously monitored by the prenatal care team without any interference or request from the researchers.”

Finally, the conclusions were drawn appropriately based on the data presented, as it was described in “Conclusion” section line 282, and below in red.

“In conclusion, the present study demonstrated that continuously assessed blood glucose levels were higher in obese pregnant women without GDM than in non-obese pregnant women and this effect was more evident at night. Additional studies may lead to the better understanding of this metabolic alteration and its possible correlation with adverse neonatal outcomes.”

2. Has the statistical analysis been performed appropriately and rigorously? Reviewer #2: No

Answer. Regarding the statistical analysis, we would like to emphasize that continuous glucose monitoring (CGM) data were analyzed using linear mixed models. Least squares means (LSM), based on the fixed terms in the model, and differences in LSM along with their 95% CIs were calculated. By thinking about changes over time, the mixed effects model for longitudinal information examination approach has the additional preferences of noticing changes more precisely by expanding the force and legitimacy of estimating the change in CGM level. Longitudinal data can be investigated utilizing different techniques, however linear mixed effect (LME) models are more appropriate in many ways. This approach is truly adaptable to represent the natural heterogeneity in the population, and can handle dropout and missing information. It additionally considers within and between wellsprings of variety. A linear mixed model is an expansion of a linear regression model for assessing longitudinal data. This statistical technique is used to assess repeated longitudinal measurements in continuous response variables in a valid and flexible manner. It tends to be utilized for information with inconsistent number of estimations per subjects. We used restricted maximum likelihood (REML) in order to obtain best less biased estimates of the covariance parameters.

3. Is the manuscript presented in an intelligible fashion and written in standard English? Reviewer #2: No

Answer: Thank you for this comment. The manuscript was previously reviewed by scientific editing service (Taylor & Francis Editing Services). The manuscript was revised again in order to correct any grammar errors.

Reviewer #3: 

The authors consider age ranges between 18-35 for OG and CG but what is the influence of age, its not mention. Groups should be uniform w. r. t. age. The hypotheses at the end of the introduction are lacking in specificity and insufficiently motivated. Please ensure that your manuscript meets PLOS ONE's style requirements, including those for file naming.

Answer: Thank you for your considerations. There is a strong positive correlation between GDM risk and maternal age (2). The exact mechanism of the association between maternal age and GDM has not been well established, but we prefer to set an age range more suitable for childbearing. A paragraph was inserted into the “Materials and methods” section, line 91, (see below highlighted in red), that clarifies this important issue. 

“Only pregnant women with gestational age between 24 to 32 weeks and aged 18 to 35 years were included. The defined age range was proposed to reduce the impact of advancing age on GDM risk.”

Moreover, a paragraph with our motivation and hypotheses was included at the end of the “Introduction “section, line 70, (see below in red highlight), which improved our manuscript.

“Although the HAPO trial have demonstrated that lower glucose values than those currently adopted to diagnose gestational diabetes resulted in improvement of adverse outcomes [13], the debate continues over what constitutes normoglycemia in pregnancy. Furthermore, there are few data on the glycemic patterns during 24 hours in obese pregnant women without GDM. Thus, the present study was designed to compare the 24-hour glycemic profile using continuous glucose monitoring of obese and non-obese pregnant women, without glucose intolerance according to the criteria proposed by the IADPSG [12]. We postulated that 24-h glucose measures (24-h area under the curve [AUC]) shall be higher among obese women than among non-obese women.”

1. Seghieri G, De Bellis A, Anichini R, Alviggi L, Franconi F, Breschi MC. Does parity increase insulin resistance during pregnancy? Diabet Med. 2005 Nov;22(11):1574-80. doi: 10.1111/j.1464-5491.2005.01693.x. 

2. Li Y, Ren X, He L, Li J, Zhang S, Chen W. Maternal age and the risk of gestational diabetes mellitus: A systematic review and meta-analysis of over 120 million participants. Diabetes Res Clin Pract. 2020 Apr;162:108044. doi: 10.1016/j.diabres.2020.108044. Epub 2020 Feb 1. PMID: 32017960.

---

## [Decision Letter · Decision Letter 1]

28 May 2021

Continuous glucose monitoring in obese pregnant women with no hyperglycemia on glucose tolerance test

PONE-D-20-32574R1

Dear Dr. Vergani,

We’re pleased to inform you that your manuscript has been judged scientifically suitable for publication and will be formally accepted for publication once it meets all outstanding technical requirements.

Kind regards,

Muhammad Sajid Hamid Akash

Academic Editor

PLOS ONE

Additional Editor Comments (optional):

Reviewers' comments:

Reviewer's Responses to Questions

**Comments to the Author**

1. If the authors have adequately addressed your comments raised in a previous round of review and you feel that this manuscript is now acceptable for publication, you may indicate that here to bypass the “Comments to the Author” section, enter your conflict of interest statement in the “Confidential to Editor” section, and submit your "Accept" recommendation.

Reviewer #1: All comments have been addressed

2. Is the manuscript technically sound, and do the data support the conclusions?

Reviewer #1: Yes

3. Has the statistical analysis been performed appropriately and rigorously? 

Reviewer #1: Yes

4. Have the authors made all data underlying the findings in their manuscript fully available?

Reviewer #1: Yes

5. Is the manuscript presented in an intelligible fashion and written in standard English?

Reviewer #1: Yes

6. Review Comments to the Author

Reviewer #1: (No Response)

7. PLOS authors have the option to publish the peer review history of their article (what does this mean?). If published, this will include your full peer review and any attached files.

Reviewer #1: No

---

## [Editor Report · Acceptance letter]

2 Jun 2021

PONE-D-20-32574R1 

Continuous glucose monitoring in obese pregnant women with no hyperglycemia on glucose tolerance test 

Dear Dr. Vergani:

I'm pleased to inform you that your manuscript has been deemed suitable for publication in PLOS ONE. Congratulations! Your manuscript is now with our production department. 

Kind regards, 

on behalf of

Dr. Muhammad Sajid Hamid Akash 

Academic Editor

PLOS ONE